# Advancing Nerve Regeneration: Translational Perspectives of Tacrolimus (FK506)

**DOI:** 10.3390/ijms241612771

**Published:** 2023-08-14

**Authors:** Simeon C. Daeschler, Konstantin Feinberg, Leila Harhaus, Ulrich Kneser, Tessa Gordon, Gregory H. Borschel

**Affiliations:** 1Department of Hand, Plastic and Reconstructive Surgery, Burn Center, Department of Plastic and Hand Surgery, University of Heidelberg, BG Trauma Hospital, D-67071 Ludwigshafen, Germany; 2Neuroscience and Mental Health Program, SickKids Research Institute, Toronto, ON M5G 1X8, Canada; 3Department of Surgery, Indiana University School of Medicine, Indianapolis, IN 46202, USA; 4Department of Surgery, University of Toronto, Toronto, ON M5G 2C4, Canada; 5Division of Plastic and Reconstructive Surgery, The Hospital for Sick Children, Toronto, ON M5G 2C4, Canada; 6Department of Ophthalmology, Indiana University School of Medicine, Indianapolis, IN 46202, USA

**Keywords:** tacrolimus, FK506, nerve regeneration, nerve injury, axon regeneration, clinical translation

## Abstract

Peripheral nerve injuries have far-reaching implications for individuals and society, leading to functional impairments, prolonged rehabilitation, and substantial socioeconomic burdens. Tacrolimus, a potent immunosuppressive drug known for its neuroregenerative properties, has emerged in experimental studies as a promising candidate to accelerate nerve fiber regeneration. This review investigates the therapeutic potential of tacrolimus by exploring the postulated mechanisms of action in relation to biological barriers to nerve injury recovery. By mapping both the preclinical and clinical evidence, the benefits and drawbacks of systemic tacrolimus administration and novel delivery systems for localized tacrolimus delivery after nerve injury are elucidated. Through synthesizing the current evidence, identifying practical barriers for clinical translation, and discussing potential strategies to overcome the translational gap, this review provides insights into the translational perspectives of tacrolimus as an adjunct therapy for nerve regeneration.

## 1. Introduction

Peripheral nerve injuries are debilitating conditions that can lead to significant functional impairment and loss of quality of life. Annually, approximately 43.8 per million people in the United States experience traumatic nerve injuries [1], predominantly affecting upper extremity nerves that are vital for hand functions [1,2]. Of those, approximately one-third require surgical nerve repair [3]. Typically, the patients are young, healthy, and economically productive [4]. For this population, an upper extremity nerve injury often translates into an inability to work for an average span of 21 to 31weeks [5,6], necessitating extended periods of rehabilitation [5,7], and in certain cases requiring a career change [4,8]. This significantly affects their mental health [9,10] and quality of life [11]. Societally, the resulting loss of productivity and potential long-term invalidity benefits could sum to over $1 million per patient over their post-injury lifetime [4], creating significant economic implications [4,5,8].

Even after meticulous microsurgical nerve repair, the functional recovery after nerve injuries often remains incomplete [6,8,12]. A significant challenge in achieving successful nerve regeneration and functional restoration lies in the inherent limited rate of axonal regeneration, averaging at most 1–2 mm/day [13,14], and the decreasing regenerative capacity of distal pathways over time when subjected to prolonged axonal deprivation [15,16,17]. Consequently, accelerating axonal regeneration may improve the functional outcomes.

One potential therapeutic approach that has gained attention in nerve regeneration research is the use of tacrolimus, a potent immunosuppressive drug with known neuroregenerative properties.

This scoping review aims to explore the translational potential of tacrolimus (FK506) for promoting nerve regeneration. Specifically, we aim to illuminate the intrinsic biological challenges and summarize the postulated mechanisms of action, concurrently reviewing both the preclinical and clinical evidence and highlighting the hurdles in clinical translation. By mapping the available evidence, identifying research gaps, and proposing strategies for clinical investigations, this scoping review seeks to provide a comprehensive overview of the translational prospects of tacrolimus for promoting the regeneration of injured nerves.

## 2. Biological Challenges to Functional Recovery after Nerve Injury

Denervation significantly affects the viability of both the denervated musculature and the motor nerves that innervate them, in a time-dependent manner. As a result, the clinical guidelines and expert consensus presently suggest a maximum denervation time of 12 to 24 months before the chance for the successful restoration of the muscle function decline to a level where most surgeons favor alternative reanimation approaches.

**Denervation atrophy**, which is characterized by the progressive remodeling of the denervated musculature, has been historically considered the main factor in poor functional recovery after peripheral nerve injuries [18,19], although recent evidence challenges this notion.

Denervated muscle fibers undergo profound cellular changes in response to denervation, including ionic imbalance [20,21], decreased resting membrane potential [22], accelerated protein catabolism [23], permeabilization of the sarcolemma [24], and activation of the intracellular inflammatome [25]. As the denervation time increases, the contractile apparatus progressively disintegrates [26,27] and the number of intramuscular mitochondria decreases [28], resulting in a substantial decline in contractility [29]. Concurrently, the intramuscular capillary bed degenerates, leading to a hypoxic intramuscular environment and fibrotic remodeling [30]. Long-term denervated muscles typically exhibit a reduced number of muscle fibers with profound atrophy, a disorganized contractile apparatus, and a surrounding dense meshwork of collagenous fibers and adipocytes [31,32]. Historically, this has been accepted as an irreversible state with permanent loss of function. However, aspects of those degenerative changes seem to be partly reversible by long-term external electrical muscle stimulation programs [31], and occasionally functional recovery has been observed even after long periods of denervation. However, attempts to reinnervate long-term denervated muscles are generally less successful. It is not that the chronically denervated muscle fibers cannot accept reinnervation. However, a key factor for limited success is that the remaining intramuscular neural pathways, which persist after the degeneration of native nerve fibers, become less conducive for regenerating nerve fibers.

**Regarding prolonged distal pathway denervation**, after nerve injury, endoneurial tubes within denervated nerve segments serve as guiding structures for regenerating axons and provide a supportive environment through proliferating Schwann cells [33,34]. In response to prolonged axonal deprivation (chronic denervation), Schwann cells atrophy [35,36] and the endoneurial tubes become progressively occluded by collagen filaments [15,16]. Collectively, these changes can reduce the number of injured motoneurons that successfully regenerate their axons within chronically denervated pathways to less than 10% [35,36]. Clinically, prolonged distal pathway denervation may occur in patients with delayed presentation after nerve injury or failed primary nerve surgery, and is usually associated with the prolonged axotomy of native nerve fibers.

**Prolonged axotomy** describes a condition in which regenerating nerve fibers are unable to establish connections with their target tissues for an extended duration. The experimental evidence indicates that chronically axotomized neurons exhibit a reduced regenerative capacity compared to freshly injured neurons. As a result, the number of injured motoneurons that regenerate successfully progressively declines as a function of time, reaching a minimum of approximately 33% compared to immediate nerve repair, even in a growth-supportive environment, i.e., following nerve transfer surgery to a freshly transected distal nerve segment [37].

Collectively, this decay of the neuronal regenerative capacity, combined with the adverse intraneural regeneration environment and the disintegration of the contractile apparatus, contributes to the low success rate of delayed reanimation approaches and proximal nerve injuries with a long regeneration distance.

## 3. Accelerating Axon Growth to Overcome Limitations in Nerve Injury Recovery

One may hypothesize that accelerating the rate of axonal regeneration decreases the denervation time of the nerves and muscles. Consequently, the axons might derive advantages from an environment that still fosters growth. This, in turn, results in an increased number of neurons establishing connections between their axons and the viable musculature within a shorter timeframe, ultimately leading to earlier and improved motor control.

The means to accelerate axonal regeneration are currently unavailable in clinical nerve surgery. Thus, surgeons have adopted nerve transfer techniques that are rooted in these principles of decreased denervation times for the nerves and muscles [38]. By converting a proximal nerve injury into a distal nerve injury by harvesting a healthy nerve of lesser importance and rerouting its axons to the very distal segment of the denervated nerve, both the recipient nerve segment and the muscle may still provide a conductive milieu and viable target.

Despite being the standard of care for proximal nerve injuries, nerve transfers inevitably result in donor side morbidity, necessitate functional relearning, and underutilize the functional capacity of the native axonal population.

Thus, there is need for innovative therapeutic strategies that can accelerate nerve regeneration and facilitate functional recovery. In this context, tacrolimus, a potent immunosuppressive drug known for its neuroregenerative properties, has emerged as a promising candidate in preclinical nerve repair studies.

## 4. Tacrolimus: A Candidate Drug for Enhancing Axonal Regeneration

### 4.1. Historical Perspective

Tacrolimus, also known as FK506, was initially isolated in 1984 from the fermentation broth of Streptomyces tsukubaensis [39], showing a potent inhibitory effect on T-cell-mediated immunity and surpassing cyclosporine’s potency by a factor of up to 100 [40]. Subsequent in vivo investigations demonstrated tacrolimus’ effectiveness in prolonging organ survival and reversing rejection across various allotransplanted organs, paving the way for its regular clinical use as an immunosuppressant to date.

The neuroregenerative properties of tacrolimus were first reported in 1994, demonstrating its capacity for promoting sensory neurite outgrowth in vitro, even at subnanomolar concentrations [41]. In the same year, Gold and colleagues observed an earlier return of muscle function and a 2.75-fold increase in the number of regenerating nerve fibers following nerve crush injuries in rats that received subcutaneous injections of tacrolimus daily (1.0 mg/kg) [42]. This work was later corroborated by a myriad of rodent nerve repair experiments, which collectively demonstrated that tacrolimus promotes axonal regeneration in vivo by 12% to 16% [43,44] and increases the number [44,45], diameter [44], and re-myelination state [44,45] of regenerating axons. These significant effects of tacrolimus translate into faster recovery of the motor function in experimental rat [42,45] and mouse [43] models.

### 4.2. Proposed Mechanisms of Action

While the precise mechanisms underlying the neurotrophic effects of tacrolimus are not fully elucidated, it is evident that the positive effects on the regrowth of regenerating nerve fibers are distinct from its immunosuppressive actions and primarily target the injured neuron. The neurotrophic effect is mediated through the FK506-binding protein (FKBP52) [46,47,48,49], which forms heterocomplexes with the 90 kDa heat shock protein (Hsp90) and its co-chaperone p23 within the neural nucleus [48]. FKBP52 plays a crucial role in guiding growth cones of regenerating neurites in response to both attractive and repulsive chemotactic signals [50], and following neuronal injury this complex undergoes redistribution to the growth cones of regenerating neurites upon exposure to tacrolimus in vitro, prompting their accelerated regeneration in vivo [48].

### 4.3. Clinical Experience

The clinical use of tacrolimus for peripheral nerve regeneration has been limited, even though it has growth-promoting effects on injured nerves. Nonetheless, there are specific clinical scenarios where its use has shown potential benefits.

After the systemic administration of tacrolimus in the context of upper-limb transplantation, remarkable rates of nerve regeneration, including the rapid progression of the Tinel sign by up to 3 mm/d and early reinnervation of intrinsic hand muscles [51,52,53], have been reported. These observations from multiple transplantation centers suggest that systemic tacrolimus administration contributes to the accelerated nerve regeneration observed in these cases. Similarly, notable functional outcomes have been observed in patients who underwent cadaveric nerve allografting for severe nerve gaps of up to 22 cm and received tacrolimus as a component of their treatment [54,55].

In a case report by Martin et al. [56], systemic tacrolimus was administered after transhumeral upper-extremity replantation in a 60-year-old woman with a poor prognosis for functional recovery. Surprisingly, the patient experienced rapid motor and sensory recovery with a 2-point discrimination of 9 mm on the thumb after 1 year and intrinsic motor reinnervation confirmed by electromyography after 19 months. In a prospective case series involving 4 patients, Phan et al. found that tacrolimus was well-tolerated [57]. However, they observed no significant improvements in sensory, motor, or functional recovery at the end of the 40-month follow-up period compared to the expected clinical outcome without treatment.

However, despite these encouraging reports, concerns over its narrow therapeutic window and systemic adverse effects associated with its systemic administration, mainly nephrotoxicity [58] and elevated levels of blood glucose [59] and liver enzymes [59], have limited its widespread application in peripheral nerve surgeries. To mitigate the systemic adverse effects of tacrolimus, researchers have directed their efforts towards developing biocompatible delivery systems that enable localized release directly at the site of nerve injury, while utilizing lower overall dosages.

### 4.4. Localized Drug Delivery: Maintaining Efficacy and Reducing Off-Target Effects

The introduction of bioengineered local delivery devices for the controlled and sustained microdosing of tacrolimus at the nerve injury site have reinvigorated the interest in leveraging its potential for peripheral nerve surgery.

Osmotic pumps have been utilized to deliver tacrolimus both systemically and locally to rats following sciatic epineural nerve repair, resulting in improved motor function compared to untreated controls [60]. However, the clinical use of osmotic pumps is hampered by their non-biodegradable nature, which increases the risk of secondary complications, including inflammation and fibrosis, and necessitates surgical removal.

Therefore, the research has evolved towards designing devices that can maintain a controllable drug delivery rate while achieving biodegradability and biocompatibility. Lin, Wang, and colleagues developed a mixed thermosensitive hydrogel (poloxamer (PLX)-poly(l-alanine-lysine with pluronic F-127)) to enable the sustained release of tacrolimus and demonstrated improved functional recovery after a sciatic nerve cut and epineural repair in mice [61,62]. While hydrogels excel in sustained drug release, precise control over the release kinetics remains challenging, potentially leading to suboptimal dosing accuracy and fluctuations in therapeutic levels due to inherent variability in the hydrogel degradation rates [63]. Tajdaran and colleagues developed a polymeric drug delivery system using poly(lactic-co-glycolic) acid (PLGA) microspheres to encapsulate tacrolimus and tested their system in a rat peripheral nerve transection and immediate epineural repair model [64,65]. The microspheres have been added to FDA-approved fibrin glue and the repaired nerve has been embedded to create a local tacrolimus-enriched environment [65,66,67]. This system has demonstrated preclinical efficacy through accelerated axonal regeneration and minimal systemic exposure to other organs. However, the translation of this system into commercial and clinical settings is impeded by the laborious preparation process at the bedside.

To address the demand for a clinically feasible and efficient approach, the research has pivoted towards the development of an off-the-shelf delivery system. Inspired by nerve wraps, which are clinically used to create a gliding bed for entrapped nerves [68,69], bioengineered tacrolimus-releasing poly(L-lactide-ε-caprolactone) (PLC) microfilms have been innovated to be wrapped around the nerve repair site in a mouse sciatic nerve transection and direct epineural repair model [70]. Recently, tacrolimus-loaded electrospun polycarbonate urethane (PCNU) nanofibers have been introduced to combine sustained release characteristics in a mesh-structured nerve wrap enabling diffusion and cell migration though the construct, thereby potentially causing less interference with the Wallerian degeneration process [71].

Rodents that were treated with these biodegradable local delivery devices as an adjunct to the gold standard epineural nerve repair demonstrated more motor and sensory neurons that regenerated their axons distal to the nerve repair site [65,70,71] and an earlier return of motor function [71]. Even in the cornea, the local and sustained release of tacrolimus has been demonstrated to accelerate ocular surface reinnervation following ophthalmic nerve injury [72].

In concomitant biodistribution and toxicity analyses, high levels of tacrolimus were detected in the regenerating nerve and its supplying spinal cord segments, supporting the hypothesis of neuronal uptake [65,71]. Conversely, the off-target levels of tacrolimus in the kidney, brain, liver, and heart at 7 and 28 days following surgery were around 80% lower compared to systemic delivery (2 mg/kg/d; *p* < 0.05) [65,71]. Consistently, local delivery resulted in a significant reduction in the peak plasma concentration if tacrolimus, reaching only 3% of the levels observed with systemic delivery [71]. Collectively, this indicates that the local release of tacrolimus may avoid toxic off-target effects while maintaining effective levels locally.

In addition to their application in direct epineural nerve repair, these local drug delivery systems hold an intriguing potential to improve functional outcomes in nerve gap reconstruction using both autologous and allogenic nerve graft repair methods [67]. However, the ideal drug delivery regimen for those interposition grafts is the subject of ongoing studies. Moreover, developing synthetic nerve guidance channels with inherent tacrolimus delivery capabilities may help to overcome the need for autologous materials for nerve gap reconstruction and substantially advance the landscape of peripheral nerve surgery [73].

## 5. The Translational Gap: Challenges in Implementing Local Tacrolimus Delivery Devices at the Bedside

The clinical implementation of local tacrolimus delivery devices represents a promising avenue for enhancing the therapeutic outcomes following nerve injury. However, historically, the transition from preclinical success to widespread bedside utilization in nerve surgery is a challenging and rarely accomplished endeavor.

This may be rooted in the fact that it is considered less attractive for big pharmaceutical companies to invest significant resources in developing treatments for nerve injuries specifically. Nerve injuries are relatively less prevalent [74,75] compared to other medical conditions, limiting the market size and potential revenue for pharmaceutical companies. Further, nerve injuries encompass a broad range of conditions with varying underlying mechanisms, locations, and severities that initially are difficult to distinguish. Each case of nerve injury may require individualized treatment and rehabilitation, involving multidisciplinary approaches over extended periods of time. These factors increase the time, costs, and risks that are associated with conducting prospective clinical trials and gaining regulatory approval, further impacting their commercial potential.

While major pharmaceutical companies may not be actively engaged in the research and development of local drug delivery systems, other stakeholders, including academic institutions, government agencies, and smaller biotech companies, may undertake efforts to commercialize such products. The successful commercialization of the Avance acellular nerve allograft (AxoGen Inc., Alachua, FL, USA) offers valuable insights in this regard. Despite its high product costs (average > $5400) [76] and niche indication for sensory nerve gap reconstructions of up to 3 cm, Avance allografts are utilized widely in clinical practice across multiple countries. The product, similar to local delivery devices for tacrolimus, originated from academic investigations in nerve repair and underwent a journey spanning over two decades before its commercial introduction [77].

In contrast to the early days of Avance, to date there are extensive clinical safety data available for local and systemically applied tacrolimus in various clinical conditions. These data can be leveraged to obtain FDA approval for clinical studies of medical combination products of tacrolimus and a biodegradable vehicle. The clinical studies may initially focus on patients undergoing primary digital nerve repair. Longitudinal assessments of tactile thresholds, two-point discrimination, pain levels, drug plasma concentrations, and device degradation via high-frequency ultrasound imaging could be conducted for safety and efficacy evaluations (Figure 1). Later studies may involve nerve transfer surgery due to its high levels of standardization compared to primary nerve repair following injury, controlled regeneration distances, and quick motor function recovery. This will allow for meaningful clinical outcome assessments within a short trial period. Two-stage procedures such as cross-face nerve grafting followed by free-functioning muscle transfer after 6–12 months may provide opportunities for distal nerve graft biopsies and detailed histomorphometric analyses of regenerated nerve fibers [78,79]. The therapeutic effect sizes could then be accurately determined for human subjects and compared with preclinical data.

## 6. Conclusions and Future Perspectives

This review outlines key biological challenges to functional recovery after nerve injury and the potential of tacrolimus as a candidate drug for improving functional outcomes. We have identified the limited rate of axonal regeneration in conjunction with a time-dependent decay of the neuronal regenerative capacity, deteriorating intraneural regeneration environment, and disintegration of the contractile apparatus as significant obstacles to successful nerve regeneration and functional restoration.

Tacrolimus, with its neuroregenerative properties, has shown promise in approaching these obstacles by promoting axonal regeneration both in preclinical studies and clinical case reports. However, its systemic administration is associated with significant adverse effects, limiting its widespread application in peripheral nerve surgeries. To address this limitation, researchers have focused on developing biocompatible, off-the-shelf delivery systems for the localized release of tacrolimus directly at the site of nerve injury. These localized delivery devices have demonstrated efficacy in promoting axonal regeneration while minimizing systemic exposure to other organs.

Despite the challenges in clinical translation, there are significant opportunities for future research and development in the field of localized tacrolimus delivery by leveraging the extensive clinical safety data. This may facilitate the regulatory process and pave the way for clinical studies that are crucial for the successful translation of devices for localized tacrolimus delivery.

In conclusion, considering the cost-effectiveness analyses on both individual and societal levels, any device that accelerates functional recovery in nerve injuries holds substantial value. While the transition from preclinical success to widespread clinical utilization has historically been challenging, the implementation of local tacrolimus delivery devices holds great promise for enhancing therapeutic outcomes in nerve injury treatments. By reducing the long-term healthcare utilization and increasing productivity and economic participation, these devices not only provide economic benefits but may also improve the mobility and overall quality of life of patients.

## Figures and Tables

**Figure 1 ijms-24-12771-f001:**
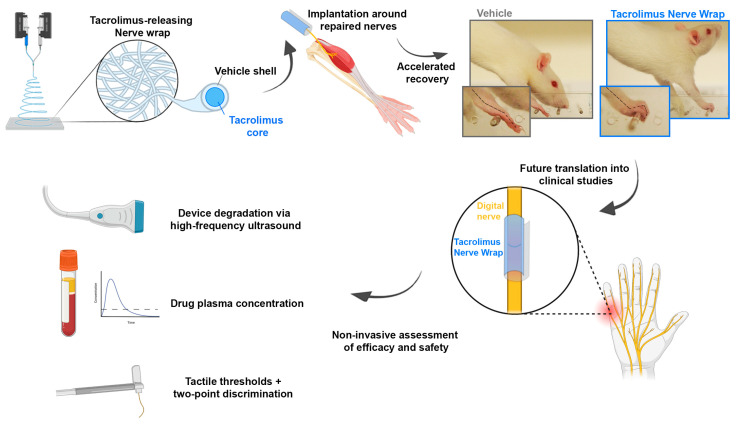
From bench to bedside—the potential clinical translation of biocompatible, off-the-shelf delivery systems for the localized release of tacrolimus directly at the site of nerve injury. Following extensive preclinical validation in vivo, we propose initial clinical studies to focus on patients undergoing primary digital nerve repair. Longitudinal assessments of tactile thresholds, two-point discrimination, pain levels, drug plasma concentrations, and device degradation via high-frequency ultrasound imaging could be conducted to evaluate the safety and efficacy.

## Data Availability

No new data were created or analyzed in this study. Data sharing is not applicable to this article.

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
