# Peer review of "Advancing Nerve Regeneration: Translational Perspectives of Tacrolimus (FK506)"

_ijms, 2023, doi:10.3390/ijms241612771_

Round 1

Reviewer 1 Report

The authors present an informative review about the promising attempt of translating local application of Tacrolismus into clinical use. They focus on the history of tacrolismus therapy for supporting peripheral nerve regeneration after surgical repair and from this they lead over to their own recent development of a tracrolismus-nerve warp that has potential to overcome shortcomings in the applicability of the therapy.

From my point of view, the work would even increase in value, when the authors add some more insights in the type of lesions investigated in the highlighted studies. This could maybe be done by adding a summarizing table or just by naming it appropriately in the text.

Furthermore, besides the fact that the tacrolismus nerve wrap seems an easy to apply of-the-shelf attempt, the authors are asked to extend their review by comparison of this approach with other recently published approaches that are currently not mentioned in their work. (e.g. tacrolismus loaden hydrogel (https://doi.org/10.3390/pharmaceutics15020508) or indirect therapy via stem cell application (https://doi.org/10.3389/fncel.2021.799151)).

Finally, the review is mainly focussing on direct nerve suture repair. However, it would be interesting to learn from the main text or the outlook, if tacrolismus therapy could also help in nerve graft repair.  The authors mention the allograft for bridging extended nerve defects. Would the authors additionally propose the development of nerve guidance channels with intrinsic tacrolismus delivery properties?

Reviewer 2 Report

The authors have submitted a well-conceived review of the literature and recommendation for advancing the use of the drug, tacrolimus (FK506), from pre-clinical studies to the clinical environment. The authors suggest leveraging extensive clinical safety data to obtain FDA approval for a combined approach of localized and temporally controlled drug delivery in a biodegradable device in patients undergoing digital nerve repair. The paper reviews the biological challenges of nerve repair, preclinical studies of tacrolimus to accelerate regeneration of severed axons, results of limited use of tacrolimus in human patients with traumatic nerve injury, and newer biodegradable devices for localized drug delivery.

As the review may have broad readership, it would be helpful to describe the principles referred to as “these” principles, which guide decisions in nerve transfer. (Line 121). Similarly, it would be informative to add an explanatory reference for the use-example of cross-face nerve graft with free functioning muscle transfer (Lines 282-285).

It would seem that the authors have focused on bioengineered devices for localized delivery of tacrolimus. They may wish to mention successes and drawbacks of other recently tested methods of drug delivery, such as hydrogel (Wang et al., 2023; Wang et al., 2022).

Round 2

Reviewer 1 Report

Congratulations to the authors! My concerns have been adequately addressed during revision.